

# Utilization of soil residual phosphorus and internal reuse of phosphorus by crops

Mei Yang and Huimin Yang

State Key Laboratory of Grassland Agro-ecosystems; Key Laboratory of Grassland Livestock Industry Innovation, Ministry of Agriculture and Rural Affairs; College of Pastoral Agriculture Science and Technology, Lanzhou University, Lanzhou, P. R. China

## ABSTRACT

Phosphorus (P) participates in various assimilatory and metabolic processes in plants. Agricultural systems are facing P deficiency in many areas worldwide, while global P demand is increasing. Pioneering efforts have made us better understand the more complete use of residual P in soils and the link connecting plant P resorption to soil P deficiency, which will help to address the challenging issue of P deficiency. We summarized the state of soil "residual P" and the mechanisms of utilizing this P pool, the possible effects of planting and tillage patterns, various fertilization management practices and phosphate-solubilizing microorganisms on the release of soil residual P and the link connecting leaf P resorption to soil P deficiency and the regulatory mechanisms of leaf P resorption. The utilization of soil residual P represents a great challenge and a good chance to manage P well in agricultural systems. In production practices, the combination of "optimal fertilization and agronomic measures" can be adopted to utilize residual P in soils. Some agricultural practices, such as reduced or no tillage, crop rotation, stubble retention and utilization of biofertilizers-phosphate-solubilizing microorganisms should greatly improve the conversion of various P forms in the soil due to changes in the balance of individual nutrients in the soil or due to improvements in the phosphatase profile and activity in the soil. Leaf P resorption makes the plant less dependent on soil P availability, which can promote the use efficiency of plant P and enhance the adaptability to P-deficient environments. This idea provides new options for helping to ameliorate the global P dilemma.

## INTRODUCTION

Phosphorus (P), a key component of nucleic acids, phospholipids and adenosine triphosphate (ATP), participates in various assimilatory and metabolic processes in plants (*Rawat et al., 2020*). Plant growth and productivity are limited by soil P availability, which ultimately affects material circulation and function of ecosystems (*Agren, Wetterstedt & Billberger, 2012*). P deficiency has adverse impacts on plant growth and productivity and may readily occur in various ecosystems especially those of agricultural systems. Around 67% of the world's agricultural land is P-deficient (*Dhillon et al., 2017*) and 51% in

Corresponding author
Huimin Yang, huimyang@lzu.edu.cn

China (*Wang, 2016*). The use of traditional manure alone can no longer sufficiently supplement P consumption in these production systems. Therefore, large amounts of P fertilizer have been applied to meet the increasing P demand. However, mined P is a nonrenewable resource. Based on the data released by the U.S. Geological Survey in 2017, statically calculated according to the P consumption rate of 2016, the verified global reserve of phosphate ore can meet the global demand of 300 years (*USGS, 2016*). For an extensive phosphate resource-consuming country, such as China, whose resource guarantee life is only 37 years according to the current production rate, the problem of P resource crisis still exists (*Zhang et al., 2017*). In addition, excess use of P fertilizer has led to the transfer of most P from terrestrial ecosystems into aquatic systems. The reimport of P back to the terrestrial ecosystems is a long and complex process with little human control. It takes nearly a million years for the P in marine sediment to participate in the P cycle of the terrestrial ecosystems again as phosphate rock (*Cordell, Drangert & White, 2009*). Therefore, to address P deficiency, it is vital to manage P fertilization efficiently and increase P use efficiency in agricultural systems.

The global population explosion and increased demands for meat and dairy products have further aggravated P consumption in the 21st century, which has led to increased severity of P deficiency in terms of global production (*Ashley, Cordell & Mavinic, 2011*; *Cordell et al., 2011*). Significant advances have been made in revealing P dynamics at the plant, ecosystem and global scales in the past few decades, with particular advancements in understanding the effects of fertilizer applications on P dynamics and the P cycle in terms of plant-soil integration (*Carpenter & Bennett, 2011*; *Venterink, 2011*). These pioneering efforts grant us chances to integrate the information and better understand more comprehensively the use of residual P in the soil and the link connecting plant P resorption to soil P deficiency, which will help in addressing the challenging issue of P deficiency. Here, we summarized the state of mechanisms related to the use of soil residual P and the internal reuse of P by plants (Fig. 1), which will help solve the contradiction between P deficiency and increased P demand (*Carpenter & Bennett, 2011*).

## SURVEY METHODOLOGY

The peer-reviewed articles in this paper were obtained from Web of Knowledge, Google Scholar, Baidu Scholar and subject-specific professional websites, scanning also in the corresponding references, selected papers, and related articles. We employed the following keywords: "phosphorus cycle", "soil residual phosphorus", "leaf phosphorus resorption", "agricultural system", "phosphate-solubilizing microorganism" and "phosphorus management". All the articles chosen in this paper should show the state of soil "residual P" and the mechanisms of utilizing this P pool, the possible effects of planting and tillage patterns and various fertilization management practices on the release of soil residual P and the link connecting leaf P resorption to soil P deficiency and the regulatory mechanisms of leaf P resorption. Both qualitative and quantitative articles were reviewed in this paper. The qualitative articles provide insights into problems by helping to understand the reasons and opinions. The quantitative articles use measurable data to express facts.

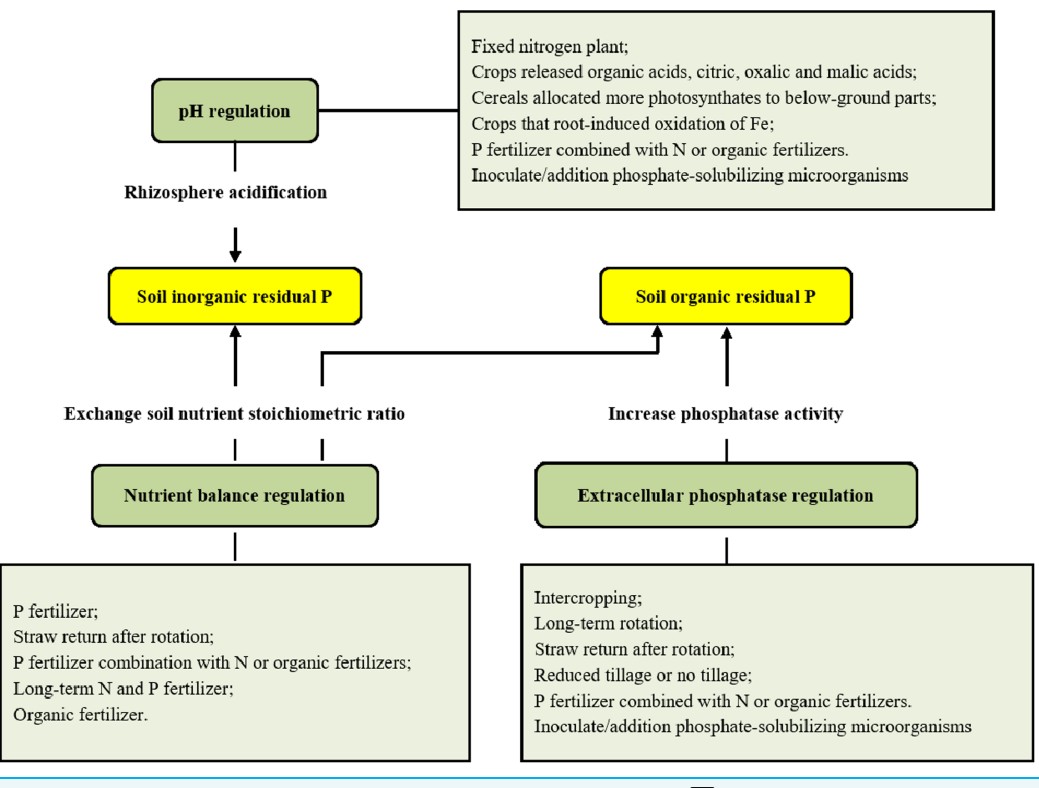

**Figure 1** **Utilization of soil residual P and its regulation.**

# UTILIZATION OF SOIL RESIDUAL P AND ITS REGULATION

## "Residual P" in soils

Generally, P in parent material enters the soil through weathering, is absorbed and stored by plants, and ultimately is returned to the soil in the form of litter (or metabolites produced by animals) (*Foster & Bhatti, 2006*). A P cycle involving utilization and transportation among plants, animals, the soil and microorganisms are formed (*Aerts & Chapin, 2000*). However, in terrestrial ecosystems such as agricultural systems, great amounts of P enter into aquatic systems, and it is difficult for this P to re-enter the effective P cycle in the short term (*Cordell, Drangert & White, 2009*). In these cases, the cycle is broken, and P deficiency can occur, seriously affecting plant growth and productivity. Therefore, P fertilizer application serves as an efficient way to supplement P, which can help to maintain and even promote the productivity of agricultural production systems (*Smit et al., 2009*). However, P is easily absorbed and settles in the form of organic and/or inorganic P with different stabilities, when soluble P components perform covalent bonds or electrostatic interactions with soil particles, or convert to insoluble forms by precipitation. And this accumulated P in the soil is difficult to convert into soluble P for plant absorption and utilization (*Sattari et al., 2012*; *Nash et al., 2014*). P fertilizer has long been overused in the farming system for decades to meet the P demand of crop growth because there is a traditional concept that the fixation of P in soils is irreversible, which

results in a large P accumulation in the soil and ineffective utilization of P (*Smit et al., 2009*). Crops absorb only 20% to 30% of inorganic orthophosphate-P in most soils after P fertilization, and the remaining P applied is rapidly fixed (*Herrera-Estrella & López-Arredondo, 2016*). *Mclaren et al. (2016)* demonstrated that the majority of the P fertilizer applied to two pasture soils in Australia was absorbed in inorganic and organic P forms, while only 35% was taken up by subterranean clover (*Trifolium subterraneum*) in the year of application. A very small portion of the P added to the soil by fertilizer, manure and/or crop stubble is used by crops the same year it is applied (*Calloway, 2016*). In contrast, a varying but often substantial portion accumulates in the soil as "residual P" (*Syers, Johnston & Curtin, 2008*), which is defined as the difference between P input (mineral fertilizer, manure, weathering, and deposition) and P output (withdrawal of P in harvested products, and P loss by runoff or erosion) in the soil (*Bouwman, Beusen & Billen, 2009*; *Sattari et al., 2012*). Soil inorganic P stocks constitute on average $1006 \pm 115$ kg ha$^{-1}$, while monoester P constitutes approximately $587 \pm 32$ kg ha$^{-1}$ (*Menezes-Blackburn et al., 2018*), indicating the enormous potential for future agronomic use.

## Utilization of inorganic residual P in soils

Fixed P in the soil can be converted into available P for plant use, and the recycling rate is up to 90% in some cases (*Smit et al., 2009*). However, there is obvious hysteresis in the use of soil residual P (*Sattari et al., 2012*). Most of the soil residual P has undergone deposition and solidification in the formation process and is ultimately fixed in different inorganic P forms. Generally, P is adsorbed mainly by oxides and hydroxides containing iron and aluminium, resulting in the formation of an iron-aluminium binding state of P in acidic soils (*Arai & Sparks, 2007*). In neutral and calcareous soils, P mostly reacts with calcium carbonate to form more stable forms such as calcium phosphate and hydroxyapatite (*Nriagu & Moore, 2012*). These forms of fixed residual P can be absorbed and utilized by plants only after they are hydrolysed into orthophosphate in the soil solution (*Ashley, Cordell & Mavinic, 2011*).

Therefore, it is fundamental in the utilization of residual P to promote the hydrolysis of soil calcium (magnesium) and/or iron (aluminium) phosphate. Existing utilization mechanisms of inorganic residual P might be divided into the following categories: (1) Direct acidification of the soil surroundings. *Arai & Sparks (2007)* found that rhizosphere acidification was an effective way to help obtain mobile P from calcareous soils. Additionally, the organic acids (like citric acid, gluconic acid, oxalic acid, and tartaric acid) (*Rawat et al., 2020*), inorganic acids (like hydrochloric acid, sulfuric acid, nitric acid, and carbonic acid) and $H_2S$ production by phosphate-solubilizing microorganisms have been reported to solubilize inorganic phosphate, although inorganic acids have low efficiency compared to organic acids (*Gaind, 2016*). (2) The cation-anion exchange balance, organic anion and proton extrusion, which lower soil pH to dissolve insoluble phosphates. *Hinsinger et al. (2003)* found that the cation–anion exchange balance and organic anion contributed to pH change in the rhizosphere since they all needed to be balanced by an exchange of charges, e.g. by the release of either $H^+$ or $OH^-$. Proton extrusion is also an alternative mode of P dissolution in the soil by microorganisms,

e.g. ammonium ($NH4^+$) (*Gaind, 2016*). (3) Root exudation and assimilation/respiration, which lead to the decrease of rhizosphere pH in neutral to alkaline soils (*Hinsinger et al., 2003*). (4) Direct redox-coupled reaction. Plant roots and associated microorganisms can also alter the pH of the rhizosphere via redox-coupled reactions, thereby improving the availability of residual P in the soils (*Hinsinger et al., 2003*). For instance, predominantly gram-negative bacteria can help the production of dominant organic acids like gluconic acid via alternative pathways for glucose oxidation and diffuse through bacterial periplasm into the surroundings (Krishnaraj & Dahale, 2014). (5) Production of siderophore by phosphate-solubilizing microorganisms (PSMs), which can be a strategy to chelate iron from Fe–P complexes in the soil. *Toscano-Verduzco et al. (2019)* found that a novel fungus, *Beauveria brongniartii*, secreted siderophores, which resulted in 59.8% of $Fe^{3+}$–Chrome azurol-S degradation and 158.95 mg $L^{-1}$ P solubilized *in vitro*. (6) Production of exopolysaccharide, which forms complexes with metal ions in the soil ($Al^{3+} > Cu^{2+} > Zn^{2+} > Fe^{3+} > Mg^{2+} > K^+$) (*Ochoa-Loza, Artiola & Maier, 2001*). This mechanism can be extrapolated as a means of P solubilization by exopolysaccharide-secreting microorganisms. Additionally, microbial cell lysis during stress conditions releases this P into the soil, which is taken up by plants and other soil organisms (*Butterly et al., 2009*).

## Utilization of organic residual P in soils

The utilization of organic P also plays a major role in the use of soil residual P. Organic P in soils generally accounts for 20–30% of the total residual P and in some cases accounts for 95% (*Rawat et al., 2020*). The vast majority of soil organic P is held by a single or double lipid bond, which can be divided into inositol (mainly from plant residue), nucleic acid (from plant and soil organism residue) and phospholipid (from plant, soil animal and microbial residue) forms (*Betencourt et al., 2012*).

The conversion from organic P to available P in soils is largely controlled by phosphatase catalysis and hydrolysis (*Kumar & Shastri, 2017*). These extracellular phosphatases produced by microorganisms and plant roots mostly belong to non-specific acidic phosphatases, and can catalyze the dephosphorylation of phosphoesters or phosphoanhydride bonds of organic compounds to promote the degradation of organic P complexes (*Sharma et al., 2013*), thereby driving the continuous P cycle between microbial fixation and release (*Richardson et al., 2011*). However, it remains unclear how much microorganisms, plant roots and other soil organisms contribute to the release of the residual P in soils (*Gaiero et al., 2018*). Generally, microorganisms (including bacteria, fungi and actinomycetes) produce acidic and alkaline phosphatases, while plants produce only acidic phosphatases (*Nannipieri et al., 2011*). For instance, fungi such as ectomycorrhizae are well known for their ability to secrete acidic phosphatases, and associated plants can thus grow well with improved P availability (*Rosling et al., 2016*). Acidic soils are dominated by acidic phosphatases, while alkaline soils are dominated by alkaline phosphatases (*Juma & Tabatabai, 1977*). The ability of alkaline phosphatases to break down substrates under alkaline conditions is obviously greater than that under neutral and acidic conditions.
Non-specific acidic phosphatases like acidic and alkaline phosphatases are typical inducible enzymes and their activity are readily affected by P supply levels. *Hofmann, Heuck & Spohn (2016)* showed that the activity of acidic and alkaline phosphatases in rhizosphere soils was 27–53% higher in P-deficient areas than in P-rich areas of *Fagus sylvatica* forests. Further, they found that at low P availability, acidic phosphatase activity increased in the rhizosphere but not in the bulk soil, and microbial phosphatase activity was not responsive to P fertilization and was still high in the P-rich soil. These results suggested that compared with microbial phosphatase, plant acidic phosphatase activity may increase more at P-deficient sites, and microbial phosphatases contributed more to total phosphatase activity than plant phosphatases in the P-rich soil (*Hofmann, Heuck & Spohn, 2016*). That is probably because phosphatase activity is regulated not only by P but also by C availability. The microbial phosphatases activity would increase to mineralize organic phosphorylated compounds extracellularly and the organic fraction of the compound can be used as a C source (*Heuck, Weig & Spohn, 2015*). In practice, microbial inoculation of plant roots can not only prevent pathogens but also induce rapid soil P cycle and reduce P fixation, which thus enhances P fertilizer utilization in agricultural systems (*Richardson et al., 2011*; *Li et al., 2018*). For instance, tomato (*Solanum lycopersicum*) under low-P conditions inoculated with *Pseudomonas* sp. RU47 primarily increase their microbial phosphatase activity in soils and stimulate the enzymatic cleavage of organic P compounds in rhizosphere and bulk soil, which promotes plant growth and utilization efficiency of P fertilizer in agricultural systems (*Nassal et al., 2018*). *Sundara, Natarajan & Hari (2002)* also reported a 25% decrease in the P requirement of sugarcane (*Saccharum hybrid*) when P fertilizer was used in combination with the P-solubilizing bacterium *Bacillus megaterium* var. *phosphaticum*.

Phytases widely found in animals, plants and microorganisms, can catalyze the removal of P from the phytate compounds (abundant organic P in soils) that is the dominant source of inositol and stored P in seeds and pollen (*Sharma et al., 2013*). Compared to plants and animals, the potential of phytase producing bacteria and fungi to obtain P from phytates is very huge (*Zineb, Trabelsi & Ayachi, 2019*). Besides, phosphatases/carbon–phosphorus (C–P) lyases can also catalyze the cleavage of the C–P bond of organophosphates, improving the P availability to plants (*Rodriguez et al., 2006*). Some studies have focused on revealing the function and mechanism of soil microorganisms involved in soil C mineralization to improve soil organic P mineralization (*Nuccio, 2014*).

## Regulation of the release of residual P in soils

Multiple different tillage and management practices are beneficial for improving the release of soil residual P in production practices to strengthen the activity of soil phosphates and acidify soil rhizosphere (Fig. 1). This will ultimately promote the use of residual P and reduce the dependence of crops on P fertilization.

### *Planting and tillage pattern*

Intercropping and crop rotation can reduce the competition for soil P between plants due to increased plant diversity, and enhance the utilization of soil original residual P and

newly accumulated organic P (*Garland et al., 2017*), mainly through more organic acid and phosphatase secretion. *Darch et al. (2018)* found that barley (*Hordeum vulgare*)/legume intercropping led to 10–70% greater P accumulation and 0–40% greater biomass than monocultures in a pot trial. The difference in the release patterns of organic acids and phosphatases by different species in the two systems may partly explain the results. More organic acid and phosphatase release can effectively increase the activation of soil residual P (*Hinsinger et al., 2011*; *Darch et al., 2018*). Compared with that under monocultures, acid phosphatase secretion under wheat (*Triticum aestivum*)/soybean (*Glycine max*) or wheat/corn (*Zea mays*) intercropping would increase, which potentially improves the use efficiency of residual P in soils (*Zhang, 2001*). In addition, soil phosphatase activity significantly increases under long-term alfalfa (*Medicago sativa*)-potato (*Solanum tuberosum*)-wheat rotation compared with fallow systems in the dryland areas on the Loess Plateau of China (*Fan & Hao, 2003*). Straw return after rotation reduces the P fixation rate in soils to improve soil P availability (*Calonego & Rosolem, 2013*). *Hallama et al. (2019)* stated that cover cropping strengthened nutrient cycling in agricultural systems under different conditions, increasing crop P nutrition and yield, as this practice could enhance soil microbial communities. Additionally, the phosphatase activity under reduced tillage is higher than that under traditional tillage (*Monreal & Bergstrom, 2000*). The promotion of phosphatase activity leads to increased release of soil residual P and an increase in soil P availability. In the rhizosphere acidification to get soil inorganic residual P, we might grow crops, e.g. white lupin (*Lupinus albus*) (*Dinkelaker, Rmheld & Marschner, 2010*) which can release organic acids, such as citric, oxalic and malic acids (*Jones et al., 2002*). The cereals allocated more photosynthates to below-ground parts, e.g. maize (*Urte et al., 2013*) or wheat, also increase root exudation and respiration to acidify soil rhizosphere (*Lambers, Atkin & Millenaar, 2002*). And the crops root-induced oxidation of Fe can decrease rhizosphere pH by redox-coupled process (*Chen, Dixon & Turner, 1980*), e.g. wetland plants and lowland rice (*Oryza sativa* L.) (*Ando, Yoshida & Nishiyama, 1983*). In general, compared with monocultures, crop rotation and intercropping significantly promote the release and utilization of residual P in soils. Although *Tang, Zhang & Yang (2015)* found that compared with tobacco (*Nicotiana tabacum*) / garlic (*Allium sativum*) intercropping, the tobacco - garlic rotation could better activate O-P, $Ca_{10}$-P and resistant organic P in soils. There are few studies comparing crop rotation and intercropping, and there is no consistent conclusion since the effects of intercropping and crop rotation on the utilization of soil residual P were different with different fertilization or co-cropped species (*Githinji et al., 2011*; *Yong et al., 2014*; *Djuniwati & Pulunggono, 2019*).

## Fertilization management

P fertilizer has long been applied to maintain high soluble P concentration in soils in production practices, although most of the P fertilizer is fixed only three hours after application (*Chang & Chu, 1961*). The combined application of P fertilizer with other fertilizers can not only improve total soil fertility but also promote the use of residual P in soils. Several reasons may explain the P availability increase in soils after mixed fertilizer application. The application of multiple inorganic fertilizers balances the individual

nutrients and each form of the same nutrient, which helps to regulate the transformation between Ca–P systems and A1-P (Fe–P) systems. This significantly increases the available inorganic P accumulation and improves the P availability in soils. Combinations with N or organic fertilizers lead to soil acidity or a change in phosphatase activity, which consequently enhances soil residual P release to improve P availability.

Long-term N and P fertilizer application causes soil acidification, which increases the available P level in calcareous soils. The increase in soil P availability under N or N plus P fertilization can also be the result of increased phosphatase activity induced by N (*Wang, Houlton & Field, 2007*; *Peñuelas et al., 2012*). Compared to the sole fertilizer applications, soil enzyme activity, microorganisms or organic P content would increase in response to applications of inorganic fertilizer combined with organic fertilizer (*Ahlgren et al., 2013*). However, P fertilizer alone hardly affects the secretion of acidic phosphatase in non-rhizosphere soils but reduces the acidic phosphatase activity in rhizosphere soils (*Spohn, Carminati & Kuzyakov, 2013*).

The application of organic fertilizers can not only increase the available P content but also improve residual P release in soils. Supply of organic matter from organic P fertilization to upland soil has been reported to decrease P-sorption and increase P desorption. This is mainly because organic and/or inorganic anion can compete with orthophosphate for the presence in the soil, leading to increase in P availability. While mineral P fertilization generally provides more reactive Al and Fe into the soil, and fixes more P (*Djuniwati & Pulunggono, 2019*). The single-lipid P content in soils increases after the application of various types of manure in practice (*Shafqat, Pierzynski & Xia, 2009*), and the combination of manure and mineral P fertilizer also significantly increases the orthophosphate portion of soil P (*Ahlgren et al., 2013*). Organic fertilizer contains a large number of highly active acidic phosphatases and soil microorganisms, and these microorganisms can also greatly increase the activity of acidic phosphatase. *Neset et al. (2008)* found that the activities of acidic phosphatase, alkaline phosphatase, phosphodiesterase and pyrophosphatase in animal waste (which can be used as organic fertilizer) were 10, 45, 50 and 160 times higher than that in soils. This activity will greatly help catalyze soil organic P in the soil, improving soil P availability. In addition, stubble is one of the most important sources of organic fertilizer in conservation agricultural systems. The return of stubble can directly increase soil P amount and help to improve residual P release. Compared with the removal of aboveground parts, the return leads to 31–63% increase in the contents of soluble P, unstable organic P and total P in soils in mixture grassland of red clover (*T. pratense*), white clover (*T. repens*), perennial ryegrass (*Lolium perenne*) and cocksfoot (*Dactylis glomerata*) (*Boitt et al., 2017*). *Zhan et al. (2015)* found that the activities of acidic, neutral and alkaline phosphatases and soil available P content with rice straw mulching were significantly higher than those without mulching. Notably, in natural systems, plant litter is the main source of organic P and acts as a stubble return to drive the P cycle (*Jiang, Yin & Wang, 2013*). In addition, phosphatase activity is also regulated by C availability in soils (*Steenbergh et al., 2011*). *Hofmann, Heuck & Spohn (2016)* found that microorganisms could use the organic part of phosphates as a C source to enhance the secretion of phosphatases. Given these advantages, organic fertilizer

application constitutes a preferred method for agricultural production to meet P requirements.

### Utilization of biofertilizer

Phosphate-solubilizing microorganisms are bioinoculants that are promising substitutes for agrochemicals, which adopts different strategies to solubilize insoluble P to soluble forms and can reduce the phosphate fertilizer input in agricultural land (*Hussain et al., 2019*). Generally, bacteria belonging to the genera *Pseudomonas, Enterobacter, Bacillus* (*Biswas et al., 2018*), *Rhizobium, Arthrobacter, Burkholderia* and *Rahnella aquatilis* HX2 (*Liu et al., 2019*), *Leclercia adecarboxylata* and fungi like *Penicillium brevicompactum* and *Aspergillus niger* (*Rojas et al., 2018*) as well as *Acremonium, Hymenella* and *Neosartorya* (*Ichriani et al., 2018*) are all potent phosphate-solubilizing microorganisms. Recently, *Astriani et al. (2020)* have discovered novel elite strains like *Pseudomonas plecoglossicida* isolated from soybean rhizosphere, which solubilized 75.39 mg $L^{-1}$ P and produced plant hormones, for instance indole acetic acid concentration was up to 38.89 ppm. These microorganisms serve as potent biofertilizers that improve the agricultural yield in harmony with ecological concerns.

In practical production, phosphate-solubilizing microorganisms are generally inoculated on the crop, added to soil, and applied together with organic/inorganic fertilizers to solubilize residual soil P to soluble forms. *Martinez et al. (2015)* found inoculation with phytate-producing bacteria like *Enterobacter* sp. N0-29PA significantly increased the biomass and P uptake of oat (*Avena sativa*) by changing the rhizosphere properties and soil enzyme activities (acidic phosphatase and urease) as well as auxin production potential without the use of fertilizer. In addition, phosphobacteria inoculation enhances the benefit of P fertilization. *Barra et al. (2019)* found the consortium of phosphobacteria (*Klebsiella* sp. RC3, *Stenotrophomonas* sp. RC5, *Klebsiella* sp. RC J4, *Serratia* sp. RC J6, and *Enterobacter* sp. RJAL6) with P fertilization improved P content in the shoot of perennial ryegrass by 29.8% compared to uninoculated control in P-deficient soils. Moreover, the addition of phosphate solubilizing bacteria such as *Bacillus, Pseudomonas, Enterobacter, Acinetobacter, Rhizobium*, and *Burkholderia* (*Teng et al., 2018*) as well as endophytic fungi such as *Aspergillus, Penicillium, Piriformospora*, and *Curvularia* (*Mehta et al., 2019*) can improve C-P lyases activity to promote the utilization of residual organic P. The application of phosphate-solubilizing microorganisms in combination with different P sources and nutrients (iron, silicon) improves P uptake and use efficiency of the crop, consequently enhancing the growth and yield of crops. *Boroumand, Behbahani & Dini (2020)* reported that phosphate-solubilizing *Pseudomonas stutzeri* and *Mesorhizobium* sp. along with nano-silica (0.05, 0.07 ppm) improved vegetative growth of land cress (*Barbarea verna*) and increased soil nitrogen and P content. This is, nano-silica might either act as a substrate for microorganisms or a stimulant that results in an increased microbial population (*Karunakaran et al., 2013*). The interaction of phosphate-solubilizing microorganisms (*Pseudomonas, Mycobacterium, Bacillus, Pantoea Rhizobia and Burkholderia*) and phosphate fertilizer improve wheat grain yield by 22% and P uptake by 26%, while reduce fertilizer input by 30%. Moreover, these

biofertilizers are safe and non-toxic to the environment (*Rawat et al., 2020*). In addition, transgenic technology has also been used in phosphate-solubilizing microorganisms to achieve efficient utilization of P resources. *Richardson (2001)* found improved P nutrition and growth in *Arabidopsis* that is genetically transformed with the *phyA* gene from *Aspergillus niger*. In recent years, metagenomics approach has also been used to modify not only phosphate-solubilizing microorganisms but also other microorganisms for improving and introducing phosphate-solubilizing efficiency (*Kumar & Shastri, 2017*).

Given the nonrenewable particularity of P resources, the utilization of mined P should be developed toward the way of "moderate mining and efficient utilization" in the future. During production, the combination of "optimal fertilization and agronomic measures" should be adopted to improve the utilization of residual P in soils to promote the use efficiency of P fertilizer and soil P stocks. A better understanding of residual P dynamics and its regulation by agricultural practices such as reduced tillage, crop rotation and stubble retention should help the conversion of various P forms in the soil, as these measures lead to changes in the balance of individual nutrients in the soil or lead to improvements in the phosphatase profile and activity in soils. The addition of organic fertilizer such as green manure also has a similar effect. Therefore, full exploration of soil residual P dynamics is still needed, and especially in conservation agricultural system, the mechanisms by which conservation practices influence soil and soil microorganisms warrant further attention. Notably, in some areas, such as the Qinghai-Tibetan Plateau of China, where extremely low temperatures have heavily restricted the release and use of soil P, the effects of cover crop change, tillage method and fertilization management also warrant deep study from the standpoint of a low-temperature background.

## INTERNAL REUSE OF P IN THE PLANT AND ITS REGULATION

### Plant P resorption

P limitation to plant growth is common in diverse ecosystems and is particularly vital for grain production due to its significant effects during the reproductive stages (*Vergeer et al., 2003*). When plants have difficulty acquiring sufficient P supplies from soils, they adjust the distribution of the limited P among various organs (tissues) to maintain metabolic activity, growth and survival. For instance, decreased litter nutrient content is often one of the nutrient preservation strategies for plants growing in N- and/or P-limited soils (*Wright & Westoby, 2003*).

Nutrient resorption refers to the process by which nutrients from senescent tissues (such as senescent leaves) are transferred to other actively growing tissues (such as green leaves and new tissues). This process can not only reallocate nutrients for the growth of new and surviving tissues of the plant (*Mao et al., 2013*), but also reduce the risk of nutrient loss with litterfall. Therefore, the plant can effectively maintain the productivity and exhibit enhanced stress resistance to soil nutrient deficiency, such as P deficiency. Retranslocation may occur in senescing leaves, stems and roots (*Gordon & Jackson, 2000*). In American desert shrubbery, where the soils contain a large number of carbonates with
strong P fixation, *Larrea tridentata* growth was not limited by soil P because there was 72–86% P resorption efficiency (*Lajtha, 1987*). *Vergutz et al. (2012)* found that approximately 64.9% of P was transferred from senesced tissues to active tissues during plant senescence at a global scale, and *Wang et al. (2014)* found that leaf P resorption efficiency of alfalfa varied with the growth stage. This efficiency can reach 67–84% at the early flowering stage of alfalfa when forage harvest is performed (*Lu et al., 2019*). Compared to N, P generally shows higher variability of resorption efficiency (proportion resorbed) and higher resorption sensitivity to nutrient availability, implying that P resorption seems more important for plant nutrient conservation and N:P stoichiometry (*Staff, 2014*). P resorption within the plant is very important for improving P use efficiency at the individual level, making the plant less dependent on soil P availability and more tolerant to soil P deficiency.

## Regulation of P resorption

Leaf P resorption differs among different genetic origins (*Sakuraba et al., 2015*) and functional groups (*Miao et al., 2019*), and is readily affected by environmental variations (*Du et al., 2017*), such as soil P availability and its balance with other nutrients (stoichiometric ratio) (*Tang et al., 2013*). However, the mechanisms to drive P resorption from senesced tissues may involve both the source-sink relationship and acidic phosphatase hydrolysis in the plant.

### Source-sink relationship

The source-sink relationship of P plays a major role in regulating P resorption of the plant. Generally, senescent tissue is the P source, while active tissues constitute the sink (*Bieleski, 1973*). When the P content in a plant exceeds the normal demand of the variety, P accumulates in senesced leaves, which causes a decrease in P resorption (*Uliassi & Ruess, 2002*). *Kobe, Lepczyk & Iyer (2005)* found that P resorption efficiency decreased with increasing P content in green leaves (P sink). Therefore, the change in nutrient concentrations in different organs (representing sources or sinks) is closely related to nutrient redistribution (*Zhang et al., 2018*). Other studies have shown that nutrient resorption efficiency is more strongly regulated by carbohydrate flux from leaves (source-sink interaction) than by factors governing the hydrolysis of nutrient-containing fractions in leaves (*Aerts, 1990*). Under P-deficient conditions, a decrease in carbohydrate accumulation is an important regulatory mechanism to enhance source-sink interaction, which primes nutrient transfer in the plant (*Demars, 1997*). *Usuda (1995)* found during mature leaf blades senescence in maize, the mature leaves were still photosynthetically active but no longer grew, thus reducing the need for P by RNA, and P from nucleic acids was therefore transferred to new leaves. Meanwhile, P deprivation may induce the early initiation and accelerate remobilization of N from old leaf blades. So soil P deficiency does not immediately slow plant growth because P can be transferred from senesced leaves to sites where P is largely needed (*Limpens, Berendse & Klees, 2003*), but leaf senescence is often accelerated by nutrient deficiency (*Bollens, 2000*). Thus, leaf senescence results in increased P resorption.

### Acidic phosphatase hydrolysis

Acidic phosphatase can hydrolyze organic P compounds in the plant into inorganic phosphate and thus help transfer phosphate from senesced tissues to young tissues. The P content accounts for 0.05–0.5% of plant dry weight (*Vance, Uhde-Stone & Allan, 2003*). Most of this P exists in plant leaves as organic P, including nucleic acids, phospholipids, and phosphorylated metabolites, among which ribosomal ribonucleic acid represents the largest organic P pool in the cell, accounting for 40–60% of the total organic P in mature leaves (*Veneklaas et al., 2012*). Plants can hydrolyze their organic P compounds into inorganic phosphate by acidic phosphatase and help transfer the phosphate from senesced tissue to young tissue. The activity of acidic phosphatase is enhanced when P is deficient in soils, and P transportation from older organs and tissues to active sites occurs sooner under low P availability. Generally, the higher the acidic phosphatase activity is in a plant, the greater the internal reuse rate of the phosphate. A recent study has shown that purple acidic phosphatases not only help activate organic P around *Arabidopsis* root systems but also promote P utilization in the plant. These phenomena are evidenced by the fact that the *AtPAP26* gene in *Arabidopsis* promotes P transfer in the leaf senescence process (*Robinson et al., 2012*). Although there are some clues that acidic phosphatase is related to P resorption and leaf senescence, the regulation of P transfer from senesced leaves is still far from fully understood (*Veneklaas et al., 2012*).

It is necessary to use the potential of a plant to maximize the P effectiveness. P resorption is an important strategy that the plant has evolved to address soil P deficiency (*Kobe, Lepczyk & Iyer, 2005*). We can use optimal species with this trait in production, e.g. evergreen tree species, to reduce the dependence on soil P and P fertilization (*Fife, Nambiar & Saur, 2008*). Although a relationship between acidic phosphatase and P resorption has been shown and the existence of acidic phosphatase genes controlling P transfer to regulate leaf senescence has also been ascertained, the metabolic network regulating P transfer in senesced leaves still needs to be revealed. A more systematic discussion of the transfer of P from senesced tissues and the control of P resorption at the molecular level should be carried out. In addition, litterfall is one of the consequences of nutrient resorption and contributes relatively large amounts of organic matter to soils, which is important in maintaining the cycles of P and other nutrients in diverse systems. However, the tradeoff of resorbed nutrients and the nutrition of litter need to be explored within individual plants. This idea may be one of the keys that unlock the resorption and residual P release.

## CONCLUSIONS

We summarized the state of soil "residual P" and the mechanisms of utilizing this P pool. The utilization can be facilitated by the acidification of the soil surroundings, the cation-anion exchange balance, organic anion and proton extrusion, root exudation and assimilation/respiration, direct redox-coupled reactions, production of siderophore and exopolysaccharide, and other pathways to utilize inorganic residual P in soils. It is also promoted by plant and microorganism secreted non-specific acidic phosphatases, phytases, and phosphatases/C-P lyases to utilize organic residual P in soils. In addition, the

phosphate-solubilizing microorganism, namely biofertilizer play an important role in the utilization of the two kinds of residual P. We also summarized the possible effects of planting and tillage patterns and various fertilization management practices on the release of soil residual P and the link connecting leaf P resorption to soil P deficiency and the regulatory mechanisms of leaf P resorption. The utilization of soil residual P represents a great challenge and a good chance to manage P well in agricultural systems. In production practices, the combination of "optimal fertilization and agronomic measures" can be adopted to utilize residual P in soils. Some agricultural practices, such as reduced or no tillage, crop rotation and stubble retention, should greatly improve the conversion of various P forms in the soil due to change in the balance of individual nutrients in the soil or due to improvements in the phosphatase profile and activity in the soil. Leaf P resorption makes the plant less dependent on soil P availability, which can promote the use efficiency of plant P and enhance the adaptability to P-deficient environments. This idea provides new options for helping to ameliorate the global P dilemma.

## ACKNOWLEDGEMENTS

We appreciated very much the help from Dr. Yuan Li, Mr. Samaila Usman and Nature Research Editing Service for language polishing. As Newton said, "If I have seen further, it is by standing on the shoulders of giants", this work would be impossible without the previous valuable research.

### Funding

This work was jointly supported by the earmarked fund for the China Agriculture Research System of MOF and MARA (CARS-34) and the National Natural Science Foundation of China (31572460). The funders had no role in study design, data collection and analysis, decision to publish, or preparation of the manuscript.

### Grant Disclosures

The following grant information was disclosed by the authors:
China Agriculture Research System of MOF and MARA (CARS-34).
National Natural Science Foundation of China: 31572460.

### Competing Interests

The authors declare that they have no competing interests.

### Author Contributions

- Mei Yang conceived and designed the experiments, performed the experiments, analyzed the data, prepared figures and/or tables, authored or reviewed drafts of the paper, and approved the final draft.
- Huimin Yang conceived and designed the experiments, performed the experiments, analyzed the data, authored or reviewed drafts of the paper, and approved the final draft.

## Data Availability

This article is a literature review without data analysis.

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
