# Peer review of "Utilization of soil residual phosphorus and internal reuse of phosphorus by crops"

_PeerJ, doi:10.7717/peerj.11704_

## Round 0.1 · original submission · Major Revisions

We have received two reviews demanding revision. Please check English again. See the remarks by reviewer #2. I suggest updating the title - put the word 'phosphorus' instead of P. PeerJ has a broad audience and the title and abstract should be easily understandable by the reader. Take more time for the update if necessary. Waiting for revised manuscript.

·

Basic reporting

This review entitled Utilization of soil residual and reuse of P within crops addressed some ideas to ameliorate the P deficiency in the soil and to optimize its use for plants.
This unreneable nutrient is essential for plant growth and productivity and its dependency has been increased worldwide.
This review brings an important discussion on the subject.

Experimental design

Some concerns
Line -173 - The authors said that the intercropping and crop rotation reduce the competition of soil P by enhancing the secretion of phosphatases. The question is: what does it mean? The phosphatases from different crops in the soil will be accumulated? Or organic P will be accumulated then mineralized?

Line 195. What would be the optimal planting? Crops rotation? Intercropping?
Organic P fertilization promotes lower amount of adsorbed P than mineral P fertilization in the soil? Why?

Line-289 Plant senescence decreases RNA, even in the grain production? Where a large amount of protein are being produced?

There are some English mistakes throughout the text.

Validity of the findings

This manuscript is a review. The discussion about the subject is very good. Some good ideas to optimize the use of P in the agriculture were presented.

I think the authors should use more current references than only old references.

Additional comments

The subject is so important. There is a lack of information and practices how to increase the P efficiency in the crop production. The demand of this essential nutrient has been increased worldwide and this review brings a great discussion.

I think the authors should use current references and answer some concerns cited above.

Reviewer 2 ·

Basic reporting

The review brings information about phosphorus (P) utilization from soil, specifically about residual P that remains adsorbed in soil particles and is not readily available for plant absorption. I think that this is a relevant subject and a review about this theme would be of great interest. The text provided by the authors brings important information, however, significant deficiencies were detected.
English language needs to be revised by professionals.
References used are most too old and more recent findings must be included.
Here I present some comments on the text, and the authors can find more details in the attached file.

Line 41: 13,19x108 hm² = 13,19x108 hm²? – This information has no reference.

Line 88: Actually, P provided by fertilization is not absorbed by soil particles, but adsorbed. The adsorption occurs when soluble P components become unavailable to the plants because they perform covalent bonds or electrostatic interactions with soil particles, or, they can convert to insoluble forms by precipitation. Several references can be included explaining this mechanism.

Line 89: What do the authors mean with “difficult to use?”

Line 100: I think it is important to state a definition for “residual P” here. The authors based much of the text in this concept but a proper definition is not presented.

Line 121; line 133: As a suggestion, it would be very interesting that the authors include more information about the solubilization of P by microorganisms. Several bacteria and fungi are able to produce organic acids, phosphatases and also phytases (not mentioned by the authors) which play essential role in the uptake of insoluble P forms to the plants. The future of “residual P” from soils for agricultural purposes certainly involves the use of microbes as main partners, thus, the theme fits in the text and would enrich the research even more. In this case, the subtitles should be reconsidered and the sections about inorganic and organic P utilization can be merged for a better understanding.

Lines 144-145: I suggest to make it clear that this enzyme is produced by the plants. The paragraph before mentions microorganisms. It can cause doubt in the way it is.

Lines 146-148: As it is stated in these lines, the reader can conclude that this enzymatic activity in the rhizosphere is only due plant metabolism, while the true is that microbes (fungi and bacteria) are also deeply involved in this process.

Lines 150-157: Several points caused doubt in these lines: 1) the authors indicate that plant phosphatases are the main enzymes involved in P uptake from P-deficient soils. Is that right? 2) If this was correctly stated, the example at the lines 155-157 is not in accordance, because it describes microbial action under low P conditions, and the authors just mentioned above that microbial action is responsible for aiding in P uptake from P-rich soils. 3) The authors can find in a generic research in scientific databases, uncountable researches describing that microbial inoculation is effective under low P conditions. In a significant frequency, microbial inoculation overcomes P uptake in relation to control treatments (with no microbial solubilizers). I think that this paragraph transmits a wrong idea and the authors should perform a deeper research, including a higher number of studies, and so it would be possible to better discuss the idea presented here.

Lines 192-193: Again, several references in this paragraph are old and more recent ones should be included to confirm the information. Specifically in these lines, crops like maize, wheat and soybean, for example, are submitted to constant crop breeding, which means that many characteristics can change. It is important to be sure that crop breeding did not affect these properties and I suggest the search for recent references which confirms this information.

Lines 213-215: The sentence has no sense. I did not understand.

Experimental design

Using the term “P cycle” to perform a research at the Web of Knowledge database, 718 results were provided, and only 8 results for “Soil residual P”. When a review is proposed, the reader expect to find important information about a subject, presented with information from the past and, especially, with the most recent findings about it. This review contains good information, but is superficial, basic, with few examples in many parts and based at most in old references. A deeper research is necessary and maybe to change the terms used for the search in the databases will be also necessary. I’ve found a great difference using “P cycle” and “Phosphorus cycle”, for example. I would also suggest other terms as “P solubilization” and “P uptake” which will bring new references that can complement ideas and provide information for a more consistent review.

Validity of the findings

Lines 331-335: The “Conclusions” section is overall redundant and the authors should improve it, merging important information and presenting their opinion in a really conclusive way. Specifically, at the mentioned lines, the authors presented referenced information, and this can not happen here. This must be included along the text in a pertinent subsection.

This is a relevant review. I encourage the authors to perform the improvement of the text to publish it.

Annotated reviews are not available for download in order to protect the identity of reviewers who chose to remain anonymous.

---

## Round 0.2 · Minor Revisions

Thanks for the manuscript update. Please fix remaining minor remarks as suggested by Reviewer #2 in the file attached. We may accept it after resubmission without additional reviewing round.

·

Basic reporting

The authors accepted suggestions and answered our concerns accordingly.

I consider that this manuscript could be accepted for publishing.

Experimental design

The study design is appropriate

Validity of the findings

The study design is appropriate

Additional comments

The authors accepted suggestions and answered our concerns accordingly.

I consider that this manuscript could be accepted for publishing.

Reviewer 2 ·

Basic reporting

The authors performed a great improvement in the text.
English language is now clear.
The review is now well structured, with good and recent literature embasing the text.
All requirements previously stated were carefully answered by the authors.
As I mentioned before, the issue of this review is of great importance nowadays, and I think that now this review will be useful for scientific research. Congratulations for the authors for thwy good job.

I've dettached in the text only some points with grammar mistakes. After correction, the paper is suitable for publication in my opinion.

Experimental design

Please see comments on Basic Report.

Validity of the findings

Please see comments on Basic Report

Annotated reviews are not available for download in order to protect the identity of reviewers who chose to remain anonymous.

---

## Round 0.3 · accepted · Accept

Thanks for the manuscript update. The reviewers have no more critical remarks. I endorse the publication.

Reviewer 2 ·

Basic reporting

The points indicated to the authors were corrected. The review is now complete and the text is suitable for publication in my opinion.

Experimental design

See Basic Report.

Validity of the findings

See Basic Report.

Additional comments

See Basic Report.